# Long-Tailed Object Detection Pre-training: Dynamic Rebalancing Contrastive Learning with Dual Reconstruction

**Chen-Long Duan[1], Yong Li[1], Xiu-Shen Wei[2]\*, Lin Zhao[1]**

[1]Nanjing University of Science and Technology
[2]School of Computer Science and Engineering, and Key Laboratory of New Generation Artificial Intelligence Technology and Its Interdisciplinary Applications, Southeast University

## Abstract

Pre-training plays a vital role in various vision tasks, such as object recognition and detection. Commonly used pre-training methods, which typically rely on randomized approaches like uniform or Gaussian distributions to initialize model parameters, often fall short when confronted with long-tailed distributions, especially in detection tasks. This is largely due to extreme data imbalance and the issue of simplicity bias. In this paper, we introduce a novel pre-training framework for object detection, called Dynamic Rebalancing Contrastive Learning with Dual Reconstruction (2DRCL). Our method builds on a Holistic-Local Contrastive Learning mechanism, which aligns pre-training with object detection by capturing both global contextual semantics and detailed local patterns. To tackle the imbalance inherent in long-tailed data, we design a dynamic rebalancing strategy that adjusts the sampling of underrepresented instances throughout the pre-training process, ensuring better representation of tail classes. Moreover, Dual Reconstruction addresses simplicity bias by enforcing a reconstruction task aligned with the self-consistency principle, specifically benefiting underrepresented tail classes. Experiments on COCO and LVIS v1.0 datasets demonstrate the effectiveness of our method, particularly in improving the mAP/AP scores for tail classes.

## 1 Introduction

With the advancement of deep learning, computer vision has seen significant progress, particularly in the development of large-scale pre-training and fine-tuning optimization paradigms [55, 59, 63, 65]. Numerous pre-training methods capture domain-specific or task-relevant concepts, boosting downstream performance [7, 14, 15, 27, 30–32, 48, 52]. In the field of object detection, current methods typically leverage ImageNet [10] and COCO [35] for pre-training, allowing partial model components, such as the backbone, to achieve satisfactory pre-training. However, these pre-training paradigms leave some key detection components randomly initialized and tend to overlook the suboptimal performance issues caused by long-tailed distributions during pre-training process.

In the traditional supervised pre-training paradigm, models are constrained by the distribution of labeled data, making it difficult for them to perform well in long-tailed settings, for example, in

---

\*Corresponding author. The first two authors contribute equally to this work. This work was supported by National Key R&D Program of China (2021YFA1001100), National Natural Science Foundation of China under Grant (62272231, 62172222), the Fundamental Research Funds for the Central Universities (4009002401), and the Big Data Computing Center of Southeast University.

tasks of pipeline failure detection [1] and face recognition [3]. While self-supervised learning has demonstrated potential in enabling models to learn richer and more effective feature representations without relying on labeled data [12, 15, 22, 27, 52], significant challenges remain. An often-overlooked but crucial challenge in long-tailed object detection is simplicity bias [23, 43, 46, 53, 54], where deep neural networks tend to rely on simpler predictive patterns while overlooking complex features that are crucial for model generalization. This bias is especially problematic for tail classes, as their limited examples make them more likely to be ignored by models that prioritize simpler patterns. To address these challenges, this work aims not only to develop a pre-training strategy that aligns with the unique demands of object detection but also to ensure its effectiveness across both balanced and long-tailed data distributions.

Motivated by this, we propose a novel pre-training framework called Dynamic Rebalancing Contrastive Learning with Dual Reconstruction (2DRCL), specifically designed for long-tailed object detection pre-training. Our method incorporates Holistic-Local Contrastive Learning, which combines holistic and local feature learning to better align the pre-training process with the fine-tuning phase. To address the issues of long-tailed distributions during pre-training, 2DRCL integrates a dynamic rebalancing strategy that improves the accuracy of tail classes. Unlike traditional resampling methods, our dynamic rebalancing sampler considers instance-level imbalance, offering more precise control over class distribution and ensuring that tail classes are adequately represented. Additionally, by introducing Dual Reconstruction, our method effectively mitigates simplicity bias, enabling the model to capture both complex patterns and nuanced features that are essential for long-tailed object detection. This dual mechanism ensures that the model not only retains detailed visual information but also grasps deeper semantic relationships, which is particularly crucial for accurately recognizing and distinguishing tail classes with limited examples.

To evaluate the effectiveness of our method, we conduct extensive experiments on two benchmark datasets, i.e., COCO [35] and LVIS v1.0 [13]. Experiments on these datasets from both quantitative and qualitative perspectives validate the effectiveness of our proposed method.

## 2  Related Work

**Pre-training for Object Detection.**    Pre-training is a critical step in object detection, often involving the use of large-scale datasets to learn transferable representations. Commonly, CNNs pre-trained on image classification datasets like ImageNet [10] are fine-tuned for object detection tasks. Self-supervised pre-training methods [4, 6, 7, 15] have gained traction in recent years. These methods do not require labeled data and aim to learn useful representations through contrastive learning. To bridge the gap between pre-training and fine-tuning, dense-level contrastive learning methods [8, 20, 28, 51, 52, 57] explored local feature similarities between views, enhancing target perception and feature learning. Recognizing the insufficiency of pre-training solely the backbone, SoCo [52] advocated pre-training additional modules like FPN to process intricate scene-level information. In object detection, methods like UP-DETR [9] and DETReg [2] pre-trained entire DETR-like detectors with region matching and feature reconstruction tasks, while AlignDet [27] froze a pre-trained backbone during detection pre-training, achieving satisfactory results with fewer epochs. Nonetheless, these approaches still struggled with effectively addressing long-tailed distribution challenges.

**Long-tailed Object Detection.**    In the literature [59, 63], repeat factor sampling [13, 58] aims to balance the data distribution by sampling tail classes more frequently. In object detection and segmentation tasks, achieving sample balance solely through straightforward resampling strategies is challenging due to the complexity of the scenes. Special loss functions represent another technical direction for tackling the long-tailed problem. EQL [45] protected tail classes from being over-suppressed by ignoring negative gradients from head samples, while EQL v2 [44] balanced gradients from head and tail classes. Seesaw loss [47] rebalanced the positive and negative gradients of each class using two reweighting factors. ECM loss [24] provided a theoretical understanding of the long-tailed tracking detection problem and introduced a novel alternative objective that optimized the margin-based binary classification error. Beyond these loss functions, methods such as supervised contrastive learning [48, 66], decoupled training [11, 40] and expert-based classifier training [56, 62, 64] have also demonstrated effectiveness under long-tailed settings. While these methods often implicitly reshape decision boundaries to protect tail classes, their indirect nature may limit their effectiveness in more complex long-tailed scenarios.

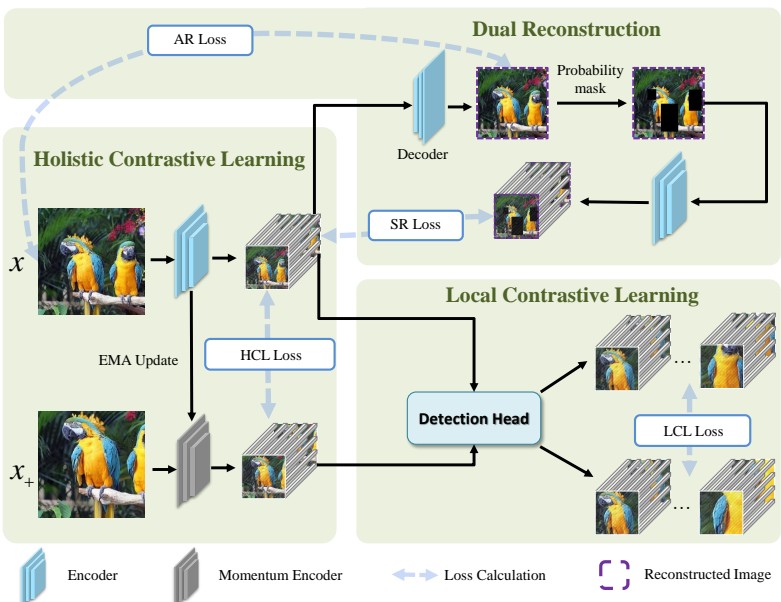

Figure 1: Illustration of the proposed Dynamic Rebalancing Contrastive Learning with Dual Reconstruction (2DRCL) method, which consists of the Holistic Contrastive Learning (Section 3.1.1), the Local Contrastive Learning (Section 3.1.2), and the Dual Reconstruction (Section 3.3). The whole network can be trained in an end-to-end manner.

## 3 Methodology

Our goal is to develop a pre-training approach tailored to the specific requirements of object detection, while maintaining robustness across both balanced and long-tailed data distributions. To this end, we introduce a novel method called Dynamic Rebalancing Contrastive Learning with Dual Reconstruction (2DRCL), specifically designed for pre-training in long-tailed object detection scenarios. In 2DRCL, we exploit a Holistic-Local Contrastive Learning (HLCL) paradigm to coordinate holistic and local feature learning to better align the pre-training with the fine-tuning phase. Building on this, a dynamic rebalancing strategy is incorporated, which emphasizes tail classes at both the image and instance (object proposal) levels to address data imbalance during pre-training. By integrating HLCL with this dynamic rebalancing strategy, we introduce a Dual Reconstruction component aimed at mitigating simplicity bias, enabling the model to concurrently capture both complex and subtle feature patterns essential for long-tailed object detection. Below, we present details of the three parts in 2DRCL.

### 3.1 Holistic-Local Contrastive Learning

In 2DRCL, the HLCL mechanism serves as the foundation for pre-training object detection models. The HLCL framework encompasses two key components: Holistic Contrastive Learning (HCL) and Local Contrastive Learning (LCL). HCL focuses on learning generic visual representations, enabling the backbone model to capture comprehensive image patterns and general semantic abstractions effectively. To integrate object-level representations into the pre-training process, LCL is introduced to guide both the backbone and the detection head toward object-level details within the image. By pre-training all network components used in object detectors, LCL ensures that the model is more precisely aligned with object detection tasks, while also enhancing its ability to capture fine-grained object-level features.

#### 3.1.1 Holistic Contrastive Learning

We present HCL mechanism in Fig. 1. As illustrated, we follow the typical CL framework, i.e., MoCo [7, 15], to realize the holistic CL in our proposed 2DRCL framework. Typically, for an image $\mathcal{I}$, we apply different image views to obtain $x$ and $x_+$ as inputs for the encoder and momentum encoder in HCL. Each view is randomly and independently augmented. Notice that the scale and location of

the same object proposal are different across the augmented image views, which enables the model to learn translation-invariant and scale-invariant object-level representations in the following LCL part, which we will elaborate next.

Subsequently, $x$ and $x_+$ are transformed via separate projectors, generating holistic-level representations, $z$ and $z_+$, which are then $\ell_2$-normalized. Subsequently, we employ the InfoNCE loss [15, 38] to drive the network training, formally:

$$\mathcal{L}_{HCL} = -\log \frac{\exp\left(z \cdot z_+/\tau\right)}{\exp\left(z \cdot z_+/\tau\right) + \sum_{i=1}^{K} \exp\left(z \cdot z_i/\tau\right)}, \tag{1}$$

where $\tau$ is a temperature hyper-parameter usually set as 0.2. For each input image, we use one positive and $K$ negative samples for HCL, where $K$ is fixed as 65,536. For the update of momentum encoder in Fig. 1, we use the same Exponential Moving Average (EMA) strategy as that in MoCo [7, 15]. Through HCL, the model is trained to effectively learn generic visual representations and capture comprehensive image patterns. However, solely relying on image-level pre-training may lead to an overemphasis on holistic representations, potentially neglecting features that are critical for object detection tasks.

### 3.1.2 Local Contrastive Learning

To introduce object-level representations into pre-training, we incorporate the LCL mechanism to bridge the gap between pre-training process and fine-tuning phase w.r.t object detection, as illustrated in Fig. 1. Specially, we employ a class-agnostic detector [26] to generate a series of proposals as bounding boxes $\mathcal{B} = \{b_1, b_2, \ldots, b_n\}$, where $b_i$ denotes the $i$-th bounding box within the augmented input image $x$. The object-level representation of a proposal is then obtained via object detection heads (e.g., RoI [17]), denoted as $z_{bb}$. The LCL loss for the local-level representation can be formulated as:

$$\mathcal{L}_{LCL} = -\log \frac{\exp\left(z_{bb} \cdot z_{bb_+}/\tau\right)}{\exp\left(z_{bb} \cdot z_{bb_+}/\tau\right) + \sum_{i=1}^{K} \exp\left(z_{bb} \cdot z_{bb_i}/\tau\right)}. \tag{2}$$

where $z_{bb_+}$ means a corresponding positive object proposal within another augmented input image $x_+$. The $K$ negative proposals means any potential proposals within other unrelated images during training. To construct a dictionary comprising a large number of object proposals from different input images, we utilize a queue-based structure. Sequences from the current mini-batch are enqueued, while the oldest mini-batch sequences are dequeued, ensuring that the dictionary size is independent of the mini-batch size. LCL mechanism maximizes the similarity between object proposals across augmented views, enabling the model to learn comprehensive representations for diverse object proposals, thus enhancing its robustness in object detection tasks.

Finally, the objective w.r.t the HLCL mechanism can be formulated as:

$$\mathcal{L}_{HLCL} = \alpha_c \mathcal{L}_{HCL} + \beta_c \mathcal{L}_{LCL}, \tag{3}$$

where $\alpha_c$ and $\beta_c$ are the weights of HCL and LCL loss, respectively.

### 3.2 Dynamic Rebalancing

To precisely control class distribution and ensure adequate representation of tail classes, we propose a dynamic resampling method that considers both images and object proposals. Unlike traditional resampling strategies, such as Repeat Factor Sampling (RFS) [13], which primarily emphasize class-balanced sampling, our approach aims to prioritize tail classes more effectively through resampling at both the image level and the object-proposal level. Given that object detection requires the identification and localization of specific objects, addressing instance-level imbalance in addition to image-level imbalance is expected to achieve a more balanced representation, particularly benefiting tail classes.

The proposed resampling method incorporates a dynamic adjustment mechanism, enabling the model to initially learn the overall distribution of the dataset and progressively shift its focus towards tail classes as pre-training advances. Specifically, for each category $c$, we calculate image-level and instance-level scores, denoted as $f_c^{im}$ and $f_c^{in}$, respectively. Here, $f_c^{im}$ indicates the proportion of images belonging to the $c$-th category in the entire dataset, while $f_c^{in}$ represents the proportion of object proposals associated with the $c$-th category across the dataset. These two scores reflect the

imbalance ratio for category $c$, following the approach used in RFS [13]. The combined score for each category, $f_c$, is then defined as the harmonic mean of these two scores:

$$f_c = \frac{f_c^{im} \cdot f_c^{in}}{\alpha_d f_c^{im} + (1 - \alpha_d) f_c^{in}} \,. \tag{4}$$

where the hyper-parameter $\alpha_d$ changes dynamically throughout pre-training, defined as $\alpha_d = \frac{T}{T_{max}}$, where $T$ is the current epoch and $T_{max}$ is the total number of pre-training epochs. As pre-training progresses, the value of $\alpha_d$ increases, gradually shifting the focus from image-level balancing to instance-level balancing, enabling the model to increasingly emphasize tail classes.

To achieve balanced sampling, we define the category-level repeat factor $r_c$ based on the score $f_c$ using the formula $r_c = \max\left(1, \sqrt{t/f_c}\right)$, where $t$ is a fixed hyper-parameter set at 0.001. This repeat factor ensures that categories with lower scores (typically tail classes) are sampled more frequently during training. The dynamic resampling strategy effectively addresses data imbalance at both the image and instance levels, enhancing focus on tail classes while mitigating the risk of overfitting due to excessive repetition of rare instances.

### 3.3 Dual Reconstruction

Building on the HLCL and dynamic resampling mechanisms, which provide the foundation for pre-training object detection and mitigate instance imbalance, respectively, our proposed 2DRCL framework introduces a Dual Reconstruction component to address simplicity bias. This component enables the model to concurrently capture both complex and subtle feature patterns, which are vital for effective long-tailed object detection.

#### 3.3.1 Simplicity Bias

Simplicity bias [23, 43, 46] is a phenomenon where models tend to favor simpler predictive patterns, often neglecting complex features that are critical for effective generalization. This issue is particularly prevalent in long-tailed distributions, where it significantly affects the performance on tail classes. In such scenarios, models struggle to learn the intricate and unique characteristics of these tail classes, further exacerbating the class imbalance problem.

To bridge this gap, we propose a Dual Reconstruction (DRC) component aimed at mitigating simplicity bias by enhancing feature discrimination for both head and tail classes. As shown in Fig. 1, DRC comprises two key elements: Appearance Reconstruction (AR) and Semantic Reconstruction (SR). The AR component enforces pixel-level reconstruction, compelling the model to capture as many subtle details as possible for each input image. In contrast, the SR component ensures semantic consistency between the features of the original input image and those of a corresponding randomly occluded image. We hypothesize that the effective implementation of DRC will enable the model to retain fine-grained visual information while also capturing deeper semantic relationships. This capability is particularly important for accurately recognizing and distinguishing tail classes, which often have limited training examples. This strategy ensures accurate visual representation while promoting a deeper semantic focus, enabling the model to better handle tail classes in long-tailed object detection.

#### 3.3.2 Appearance Reconstruction

To enforce appearance consistency, we utilize an auto-encoding structure specifically designed for high-fidelity reconstruction of input images. The encoder $f$, parameterized by $\theta$, maps an input image $x$ into a dense feature space, represented as $z = f(x)$. A generator $g$, parameterized by $\eta$, then attempts to invert this mapping, producing a reconstructed version: $\hat{x} = g(f(x))$. Through pixel-wise image reconstruction, the Appearance Reconstruction (AR) component compels the latent features, $f(x)$, to capture as many subtle details as possible for each input image.

AR is not merely replicating the input image; rather, it acts as an auxiliary regularization mechanism that focuses on distilling discriminative visual features relevant to the primary object detection task for each input image. By enforcing image reconstruction, AR enables the model to effectively capture both prominent and nuanced details present in the input data. The AR loss is formulated as a

pixel-wise mean-squared error (MSE), expressed as:

$$\mathcal{L}_{AR} = \|x - g\left(f\left(x\right)\right)\|_2^2 \ . \tag{5}$$

### 3.3.3 Semantic Reconstruction

While AR ensures that the model captures fine-grained visual details essential for accurately representing and distinguishing between different objects, especially in cases with limited examples of tail classes, it is equally important to maintain semantic integrity in the reconstructed images. This semantic consistency allows the model to focus on the underlying meaning and context of the image, rather than merely surface-level details, thereby promoting a more robust and generalized understanding of each input.

To address this need, we introduce Semantic Reconstruction (SR), which incorporates controlled perturbations during the reconstruction process. SR is designed to preserve the semantic content of the original image while allowing the model to learn to recognize and reconstruct meaningful features even when certain parts of the image are altered or obscured. This approach ensures that the model develops a deeper understanding of each input's inherent structure and context.

Specifically, we apply a mask to a fixed percentage (e.g., 25%) of an object proposal within the reconstructed image $g\left(f\left(x\right)\right)$, resulting in a masked version, denoted as $\mathcal{M}\left(g\left(f\left(x\right)\right)\right)$, where $\mathcal{M}$ means image masking operation. This masked image is then re-encoded by the encoder to generate the corresponding latent features, $\hat{z} = f\left(\mathcal{M}\left(g\left(f\left(x\right)\right)\right)\right)$. The Semantic Reconstruction (SR) loss is computed by measuring the congruence between the feature representations of the vanilla images and those of the masked reconstructed images, evaluated across multiple layers of the network. This approach ensures that the model maintains semantic consistency while learning to recognize and reconstruct meaningful features.

$$\mathcal{L}_{SR} = \sum_{p=1}^{P} \|f\left(x\right) - f\left(\mathcal{M}\left(g\left(f\left(x\right)\right)\right)\right)\|_2^2 \ , \tag{6}$$

where $P$ represents the number of feature layers considered, and the SR loss, $\mathcal{L}_{SR}$, is defined as the Euclidean distance between the original (vanilla) features and the reconstructed features across these layers. The SR component ensures that even in the presence of visual disruptions, the essential semantic features are preserved, allowing the model to learn robust, invariant features that go beyond superficial visual similarities. This approach enhances the model's ability to generalize by focusing on meaningful semantic information rather than just appearance.

Conclusively, our proposed DRC leverages both appearance and semantic consistency to address simplicity bias, encouraging the model to learn rich and complex feature representations essential for accurate and robust detection of tail classes. The interplay between the two reconstruction losses enhances the model's sensitivity to both fundamental visual details and higher-level semantic features, leading to a more versatile and effective detection paradigm. This combined approach ensures that the model not only captures detailed visual information but also grasps abstract semantic relationships, improving its overall performance in long-tailed object detection tasks.

The total loss for the Dual Reconstruction combines the AR and SR losses can be formulated as:

$$\mathcal{L}_{DRC} = \alpha_r \mathcal{L}_{AR} + \left(1 - \alpha_r\right) \mathcal{L}_{SR} \ , \tag{7}$$

where $\alpha_r$ balances the trade-off between visual fidelity and semantic accuracy. This dual-focus strategy force the model to reconstruct the image/features for both the head and the tail classes. This, DRC enhances the model's ability to represent and detect tail classes effectively.

Overall, the final loss function of our method is optimized by:

$$\mathcal{L} = \mathcal{L}_{HLCL} + \mathcal{L}_{DRC} + \mathcal{L}_{det} \ , \tag{8}$$

where $\mathcal{L}_{det}$ denotes the loss of object detection that makes the pre-training consistent with the task. For simplicity, the weights of all losses in $\mathcal{L}$ are set to 1.

## 4 Experiments

In this section, we outline the experimental settings, implementation details, and main results. Additionally, a comprehensive description of the experimental settings is provided in Section A.1 of the Appendix.

Table 1: Comparisons with state-of-the-art methods on COCO (Mask R-CNN with R50-FPN).

| Backbone Initialization | Methods | $AP^{bb}$ | $AP^{bb}_{50}$ | $AP^{bb}_{75}$ | $AP^{mk}$ | $AP^{mk}_{50}$ | $AP^{mk}_{75}$ |
|---|---|---|---|---|---|---|---|
| From scratch | DenseCL [51] | 39.6 | 59.3 | 43.3 | - | - | - |
| | Self-EMD [36] | 40.4 | 61.1 | 43.7 | 37.4 | 56.5 | **39.7** |
| | SoCo [52] | 40.6 | 61.1 | 44.4 | - | - | - |
| | SlotCon [57] | 41.0 | 61.1 | 45.0 | - | - | - |
| ImageNet pre-trained backbone | Surpervised | 38.3 | 58.0 | 42.1 | 34.3 | 54.9 | 36.6 |
| | AlignDet [27] | 39.4 | 59.2 | 43.2 | 35.3 | 56.1 | 37.7 |
| | Ours | **41.4** | **61.3** | **45.8** | **37.4** | **57.2** | 39.4 |

## 4.1 Experimental Configurations

**Datasets.** We conduct experiments on two representative datasets: COCO [35] and LVIS v1.0 [13]. The COCO dataset is a standard benchmark for object detection, segmentation, and captioning tasks, comprising 80 classes with a relatively balanced distribution, including 118k training images and 5k validation images. Given the balanced nature of the class distribution in COCO, we use this dataset to evaluate the performance of the proposed 2DRCL under balanced settings. In addition, we utilize the LVIS v1.0 dataset to benchmark long-tailed object detection scenarios. LVIS features 1,203 classes with a highly imbalanced distribution, containing 100k training images and 19.8k validation images. The classes in LVIS are categorized into three groups based on their frequency of occurrence [13]: rare (1~10 instances), common (11~100 instances), and frequent (>100 instances). This categorization allows for a comprehensive assessment of 2DRCL's performance under long-tailed data distributions.

**Implementation Details.** Experiments are conducted with both Faster R-CNN and Mask R-CNN frameworks. For a comprehensive comparison, we use both ResNet-50 and ResNet-101 backbones. All models are implemented using the MMDetection toolbox [5]. We pre-train the models on 8 RTX3090 GPUs with a batch size of 16. Unless otherwise specified, pre-training follows the $1\times$ schedule (12 epochs), starting with an initial learning rate of 0.02, which is reduced by a factor of 10 after the 8th and 11th epochs. For $2\times$ schedule, models are trained with 24 epochs, and the learning rate decays at the end of epoch 16 and 22. In our experiments, the hyper-parameters are set as follows: $\alpha_c$ is set to 0.1, $\beta_c$ is set to 0.05, $\alpha_r$ is set to 0.1. When conducting experimental comparisons on the LVIS v1.0 dataset in Table 3, we first use our 2DRCL for pre-training, followed by the application of existing long-tailed methods for fine-tuning to further enhance performance. Finally, we select 'ECM [24]+2DRCL' as 'Ours' for comparison with state-of-the-art methods.

## 4.2 Quantitative Results

**Mask R-CNN with R50-FPN on COCO dataset.**
Table 1 presents the comparison, where all methods are pre-trained on the COCO training dataset and evaluated on the COCO validation dataset. Typically, the backbone can be initialized either from scratch or using an ImageNet pre-trained model. Methods such as DenseCL [51], Self-EMD [36], and SoCo [52], which are initialized from scratch, achieve an $AP^{bb}$ ranging from 39.6% to 41.0%. Notably, these methods rely on 200~800 epochs for training. As a comparison, methods that utilize an ImageNet pre-trained style, including AlignDet and our proposed 2DRCL, require only 12 epochs for pre-training. Among the compared methods in Table 1, our 2DRCL demonstrates superior object

Table 2: Comparisons with pre-trained methods on LVIS v1.0 with a $1\times$ scheduler using Mask R-CNN.

| Method | $AP^{bb}$ | $AP^{bb}_r$ | $AP^{bb}_c$ | $AP^{bb}_f$ |
|---|---|---|---|---|
| MoCo v2 [7] | 14.5 | 3.9 | 12.4 | 21.6 |
| SimCLR [6] | 19.9 | 8.0 | 18.1 | 27.1 |
| BYOL [12] | 15.3 | 5.4 | 13.2 | 21.9 |
| SoCo [52] | 17.6 | 5.3 | 15.9 | 24.9 |
| AlignDet [27] | 22.6 | 10.3 | 20.8 | 29.9 |
| Ours | **23.9** | **11.9** | **22.3** | **31.0** |

detection performance, achieving the highest $AP^{bb}$ of 41.4% and $AP^{mk}$ of 37.3%, significantly outperforming both AlignDet and the supervised baseline. This improvement can be attributed to 2DRCL's capability to narrow the gap between pre-training and fine-tuning. By effectively bridging this gap, 2DRCL is able to leverage the benefits of the pre-trained model more efficiently for object detection tasks.

Table 3: Comparisons with state-of-the-art methods on LVIS v1.0 with a $2\times$ schedule.

(a) Faster R-CNN with R50-FPN.

| Method | $AP^{bb}$ | $AP^{bb}_r$ | $AP^{bb}_c$ | $AP^{bb}_f$ |
|---|---|---|---|---|
| BCE [42] | 19.5 | 1.6 | 16.6 | 30.6 |
| RFS [13] | 24.2 | 14.2 | 22.3 | 30.6 |
| DropLoss [21] | 21.8 | 5.2 | 21.8 | 29.1 |
| PCB [19] | 23.0 | 6.2 | 21.5 | 32.2 |
| EQLv2 [44] | 25.4 | 15.8 | 23.5 | 31.7 |
| Seesaw [47] | 26.4 | 16.8 | 25.1 | 32.2 |
| BAGS [29] | 23.7 | 14.2 | 22.2 | 29.6 |
| ACSL [50] | 22.2 | 9.9 | 21.3 | 28.5 |
| LOCE [11] | 25.1 | 15.7 | 24.2 | 30.1 |
| BACL [40] | 26.1 | 16.0 | 25.7 | 30.9 |
| ECM [24] | 26.7 | 17.5 | 25.7 | 32.2 |
| Ours | **27.3** | **18.6** | **25.8** | **32.6** |

(b) Mask R-CNN with ResNet-50/101.

| Backbone | Method | AP | $AP_r$ | $AP_c$ | $AP_f$ | $AP^{bb}$ |
|---|---|---|---|---|---|---|
| R50-FPN | CE | 18.7 | 0.4 | 16.5 | 29.3 | 19.7 |
| | RFS [13] | 23.7 | 14.2 | 22.9 | 29.3 | 24.7 |
| | EQLv2 [44] | 25.2 | 17.4 | 24.1 | 29.9 | 26.0 |
| | LOCE [11] | 26.6 | 18.5 | 26.2 | 30.7 | 27.4 |
| | SeeSaw [47] | 26.9 | 19.6 | 26.8 | 30.5 | 27.3 |
| | ECM [24] | 27.4 | 19.7 | 27.0 | 31.1 | 27.9 |
| | Ours | **27.7** | **20.4** | **27.1** | **31.4** | **28.3** |
| R101-FPN | CE | 25.5 | 16.6 | 24.5 | 30.6 | 26.6 |
| | EQLv2 [44] | 27.2 | 20.6 | 25.9 | 31.4 | 27.9 |
| | SeeSaw [47] | 28.2 | 20.3 | 28.1 | 31.8 | 29.0 |
| | ECM [24] | 28.7 | **21.9** | 28.4 | 32.2 | 29.4 |
| | Ours | **28.8** | 21.1 | **28.7** | 32.3 | **29.6** |

**Comparisons with Pre-trained Methods on LVIS v1.0.** In Table 2, we present a comparison of our method with several state-of-the-art pre-trained methods on the LVIS v1.0 dataset using the Mask R-CNN framework with a $1\times$ scheduler. The results obviously illustrate the limitations of existing pre-training approaches in addressing the challenges posed by long-tailed distributions w.r.t the object detection tasks. Specifically, traditional pre-training methods consistently demonstrate inferior performance on tail classes, as evidenced by their relatively low $AP^{bb}_r$ scores. For instance, MoCo v2 [7] and BYOL [12] achieve $AP^{bb}_r$ scores of 3.9% and 5.3%, respectively, indicating a obvious deficiency in precisely detect the target objects w.r.t the long-tailed classes. Our proposed 2DRCL is specially designed for long-tailed object detection pre-training and shows consistent superiority in Table 2. By dynamically rebalancing the data distribution and incorporating Dual Reconstruction mechanisms, 2DRCL effectively captures object-level characteristics for both head and tail classes. The superior performance of the proposed 2DRCL highlights its efficacy in addressing long-tailed object detection challenges, demonstrating a strong capability to focus on tail classes and alleviate the inherent imbalance issues in such datasets.

**Comparisons with State-of-the-art Long-tailed Object Detection Methods on LVIS v1.0.** To evaluate the effectiveness of our method for long-tailed object detection, we compare 2DRCL with state-of-the-art techniques across different object detection frameworks (Faster R-CNN and Mask R-CNN) and backbone networks (ResNet-50 and ResNet-101) on the LVIS v1.0 dataset. As shown in Table 3, our method achieves the highest accuracy in both $AP^{bb}$ and AP. Specifically, for the Faster R-CNN framework, our pre-training technique outperforms all the competitors, particularly in $AP^{bb}$ and $AP^{bb}_r$. This advantage is consistently observed with the Mask R-CNN framework as well. We attribute this improved performance, particularly for tail classes, to 2DRCL's dynamic rebalancing of the data distribution and the introduction of Dual Reconstruction mechanisms. The effectiveness of 2DRCL stems from its ability to significantly mitigate extreme imbalance and simplicity bias for tail classes during the pre-training phase. We will investigate the contribution for each of components in 2DRCL in Section 4.2.

**Discussions.** To address concerns that the performance gains might be attributed to longer training durations, we evaluate the COCO dataset using various fine-tuning schedules, with the results presented in Table 4a. The findings demonstrate that even with $1\times$ fine-tuning (12 epochs), our method surpasses the baseline trained with $4\times$ fine-tuning, indicating that the observed performance improvements of 2DRCL are not merely due to extended training epochs. Furthermore, Table 4b shows results on the LVIS v1.0 dataset, where all methods are compared under the same total number of training epochs for a fair evaluation. Specifically, while long-tailed methods are fine-tuned for 12 epochs, our 2DRCL employs 6 epochs pre-training followed by 6 epochs fine-tuning, maintaining an equivalent overall training duration. The results reveal an average improvement of 0.5% in $AP^{bb}$ and 1.3% in $AP^{bb}_r$ with 2DRCL's pre-training strategy, underscoring the effectiveness of 2DRCL in enhancing long-tailed object detection performance.

Table 4: Comparisons w.r.t different training/fine-tuning epochs.

(a) Comparisons under different fine-tuning epochs on COCO. The preceding four methods exploit ImageNet pre-trained backbone.

| Fine-tuning Schedule | $AP^{bb}$ | $AP^{bb}_{50}$ | $AP^{bb}_{75}$ |
|---|---|---|---|
| $1\times$ | 38.3 | 58.0 | 42.1 |
| $2\times$ | 38.8 | 58.4 | 42.4 |
| $3\times$ | 39.0 | 58.7 | 42.9 |
| $4\times$ | 39.2 | 59.5 | 42.9 |
| $1\times$ (Ours) | 41.4 | 61.3 | 45.8 |

(b) Results on LVIS v1.0 with same training epochs.

| Methods | $AP^{bb}$ | $AP^{bb}_r$ | $AP^{bb}_c$ | $AP^{bb}_f$ |
|---|---|---|---|---|
| RFS | 22.7 | 9.1 | 21.5 | 30.0 |
| +Ours | **23.3** | **10.3** | **21.7** | **30.3** |
| EQL | 24.9 | 14.8 | 24.1 | **30.4** |
| +Ours | **25.2** | **15.9** | **24.3** | 30.3 |
| Seesaw | 24.7 | 14.7 | 23.6 | 30.4 |
| +Ours | **25.2** | **15.6** | **24.2** | **30.5** |
| ECM | 26.5 | 17.0 | 25.4 | 31.7 |
| +Ours | **27.0** | **19.0** | **25.8** | **31.9** |

**Ablation Analysis across 2DRCL Components.** To investigate the contribution for each component in 2DRCL, we evaluate our 2DRCL on the LVIS v1.0 dataset. As shown in Table 5, incorporating HLCL, which combines HCL and LCL, results in a 1.7% improvement in $AP^{bb}_r$ over the baseline. HCL focuses on learning generic visual representations, enabling the backbone to capture comprehensive image patterns and semantic abstractions, while LCL ensures precise alignment with object detection tasks and enhances the capture of fine-grained object-level features. Additionally, DRB dynami-

Table 5: Ablations for various components in our 2DRCL on LVIS v1.0.

| HCL | LCL | DRB | AR | SR | $AP^{bb}$ | $AP^{bb}_r$ | $AP^{bb}_c$ | $AP^{bb}_f$ |
|---|---|---|---|---|---|---|---|---|
| ✗ | ✗ | ✗ | ✗ | ✗ | 22.7 | 9.1 | 21.5 | 30.0 |
| ✓ | ✗ | ✗ | ✗ | ✗ | 22.5 | 10.5 | 21.0 | 29.3 |
| ✗ | ✓ | ✗ | ✗ | ✗ | 21.9 | 9.8 | 20.8 | 28.7 |
| ✓ | ✓ | ✗ | ✗ | ✗ | 22.4 | 10.8 | 21.1 | 29.0 |
| ✓ | ✓ | ✓ | ✗ | ✗ | 23.8 | 14.3 | 22.3 | 30.1 |
| ✓ | ✓ | ✓ | ✓ | ✗ | 24.2 | 14.9 | 22.6 | 30.3 |
| ✓ | ✓ | ✓ | ✓ | ✓ | **24.4** | **15.2** | **22.7** | **30.3** |

cally rebalances the data distribution and helps to significantly boosts performance for tail classes, leading to a 3.5% improvement in $AP_r^{bb}$. The Dual Reconstruction (DRC) mechanism, comprising AR and SR, brings the total improvement to 6.1%. AR enforces pixel-level reconstruction, compelling the model to capture subtle visual details, while SR ensures semantic consistency between the original and occluded images. This combination allows the model to retain intricate visual information while capturing deeper semantic relationships, resulting in enriched and coherent feature representations.

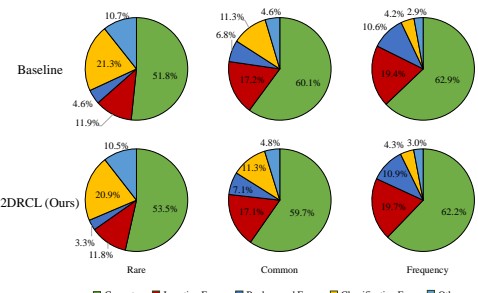

Figure 2: Error analyses comparisons. 2DRCL achieves superior performance on tail classes without significantly compromising accuracy for the more frequent classes.

## 4.3 Further Analysis

In this section, we conduct a thorough analysis of our proposed 2DRCL, emphasizing its role in mitigating simplicity bias and enhancing feature representation through DRC mechanism.

**Error Analyses.** To determine which error types our 2DRCL method effectively mitigates, we conducted an error analysis experiment. Following the error categorization paradigm from YOLO [41], we classify the top N predictions for each class into five error types. The pie charts in Figure 2 show the distribution of these errors for rare, common, and frequent classes on the LVIS v1.0 validation set. As shown in Figure 2, our 2DRCL shows noticeable improvements for rare object classes,

with correct predictions increasing from 51.8% in the baseline to 53.5%, alongside a reduction in both non-background classification errors and background prediction errors. This suggests that our 2DRCL enhances the model's ability to accurately classify the rare objects and accurately distinguish them from the background. Although there is a slight accuracy decrease for common and frequent classes, this trade-off is minimal, with the gains in rare class detection outweighing these minor losses. This demonstrates that our method effectively addresses long-tailed object detection challenges by improving performance on tail classes without obviously compromising accuracy for other frequent classes across the dataset.

**Simplicity Bias Analyses.**    To explicitly illustrate how our method addresses simplicity bias, we present a visualization of the activations corresponding to randomly sampled test images from the LVIS v1.0 dataset in Figure 3. The results demonstrate that 2DRCL effectively mitigates simplicity bias in long-tailed object detection by learning more comprehensive patterns that encompass informative regions, particularly for images belonging to tail classes. In comparison, 2DRCL consistently identifies more critical regions than ECM [24], highlighting the superiority of our approach in addressing simplicity bias. The comparisons presented in the fourth and fifth rows underscore the effectiveness of the proposed DRC mechanism, revealing that the introduction of the DRC mechanism significantly enhances feature attention on tail classes while reducing background interference. This finding further indicates that the DRC plays a crucial role in mitigating simplicity bias, enabling the model to retain intricate visual details and capture deeper semantic relationships, thereby producing enriched and coherent feature representations.

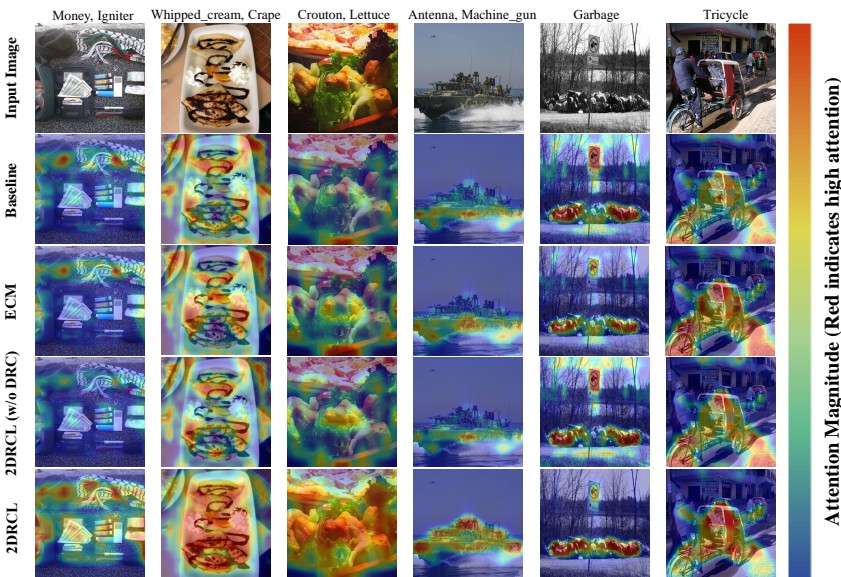

Figure 3: Attention map comparisons w.r.t Baseline [13], ECM [24], 2DRCL (w/o DRC) and 2DRCL (our method) on LVIS dataset. The top row shows the corresponding class names of the input images. Best viewed in color.

## 5   Conclusions and Limitations

We proposed Dynamic Rebalancing Contrastive Learning with Dual Reconstruction (2DRCL) to address the challenges posed by long-tailed distributions in object detection pre-training. By integrating holistic and local contrastive learning with dynamic rebalancing and dual reconstruction, 2DRCL aligned the pre-training strategy with the specific demands of object detection, ensuring effectiveness across both balanced and long-tailed data. It successfully mitigated simplicity bias for tail classes, enhancing their feature representations and overall performance. Experiments demonstrated significant improvements in attention to tail classes and reduced background errors, as confirmed by both quantitative and qualitative analyses. However, our method had limitations, particularly in its relatively high computational costs. Future work will focus on optimizing computational efficiency.

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

# A  Appendix / supplemental material

In the supplementary materials, we present further information about the proposed 2DRCL pre-training framework, including: 1) More detailed experimental settings, including the specifics of pre-training and downstream fine-tuning, as well as the setup for error analysis; 2) Additional experimental results for further analysis.

## A.1  Implementation Details

**Pre-training Settings.**  First, we generate a series of high-quality bounding boxes using a class-agnostic detector [26]. Then, we randomly select 8 bounding boxes from this set for subsequent pre-training. Through the introduction of object proposals, the architectural discrepancy is reduced between pre-training and downstream detection fine-tuning. Faster R-CNN [42] and Mask R-CNN [16] are commonly adopted frameworks to evaluate transfer performance. We employ MMDetection [5] as our detection framework to conduct our experiment. Both the projection network and prediction network are 2-layer MLPs which consist of a linear layer with output size 256 followed by batch normalization [25], rectified linear units (ReLU) [37], and a final linear layer with output dimension 256. Once all views are constructed, we employ the data augmentation pipeline of MoCo [7, 15]. Our generator architecture consists of four deconvolutional layers with dimensions (2048, 512), (512, 256), (256, 64), and (64, 3), respectively, and each layer uses a kernel size of 4. ReLU is used for non-linear activation between the layers. Specifically, we apply random horizontal flip, random crop, color distortion, Gaussian blur, and the solarization operation. The models are trained with a total batch size of 16 on 8 GPUs (RTX3090 with 24 GB VRAM). Unless otherwise specified, all pre-training follows the default $1\times$ (12 epochs) schedule. In each stage, the learning rate starts at 0.02 and decreases by 0.1 after 8 and 11 epochs, respectively. If not specified, the supervised pre-trained ResNet [18] in PyTorch [39] is used by default for both the pre-training and fine-tuning stages.

**Training Details.**  We reproduce multiple methods with different paradigms as our baselines, including end-to-end and decoupled methods, such as RFS [13], SeeSaw [47], ECM [24], ROG [60], LOCE [11] and BACL [40], following their default experiment settings. In terms of the model architecture, we opt for the popular ResNet [18] with FPN [33] as the backbone and train detection models of Faster-RCNN and Mask-RCNN for $1\times$ or $2\times$ scheduler. We trained the models using SGD with 0.9 momentum. The batch size and learning rate are set as 16 and 0.02, and the data augmentation strictly follows previous long-tailed detection methods [24, 47]. For $1\times$ schedule with 12 training epochs, the learning rate is initialized as 0.02, and then decays by 0.1 at the end of epoch 8 and 11. For $2\times$ schedule, models are trained with 24 epochs, and the learning rate decays at the end of epoch 16 and 22. We evaluated our models using both COCO and LVIS metrics. For COCO, we report object detection metrics including average precision for bounding boxes ($AP^{bb}$), AP with an IoU threshold of 50% ($AP^{bb}_{50}$), and AP with an IoU threshold of 75% ($AP^{bb}_{75}$). For instance segmentation, we report $AP^{mk}$ (AP for masks), $AP^{mk}_{50}$, and $AP^{mk}_{75}$. The LVIS evaluation includes mean average precision (mAP), AP at an IoU of 50% ($AP_{50}$), AP at an IoU of 75% ($AP_{75}$), as well as AP for rare ($AP_r$), common ($AP_c$), and frequent classes ($AP_f$). For Mask R-CNN, we report AP for instance segmentation and $AP^{bb}$ for object detection.

**The Setting of Error Analyses.**  Following the settings of [41], we choose the top N predictions for each category during inference time. Each prediction is classified based on the type of error:

- Correct: correct class and IOU > 0.5
- Location Error: correct class and 0.1 < IOU < 0.5
- Background Error: IOU < 0.1 for any object
- Classification Error: class is wrong and IOU > 0.5
- Other: class is wrong and 0.1 < IOU < 0.5

## A.2  Additional Experiment

**Consistent Improvements.**  We evaluate the effectiveness of our method on the LVIS v1.0 dataset by combing the proposed 2DRCL method with existing long-tailed object detection methods. As

Table A.1: Experiments on **LVIS v1.0**. We combine eight existing methods with our method '2DRCL'. The ResNet-50-FPN and ResNet-101-FPN are adopted as backbones for Mask R-CNN. We reproduced all methods using their official code and trained with a $1\times$ schedule, totaling 12 epochs.

| Strategy | Schedules | Methods | +Ours | LVIS v1.0 (ResNet-50-FPN) | | | | LVIS v1.0 (ResNet-101-FPN) | | | |
| --- | --- | --- | --- | --- | --- | --- | --- | --- | --- | --- | --- |
| | | | | $AP^{bb}$ | $AP^{bb}_r$ | $AP^{bb}_c$ | $AP^{bb}_f$ | $AP^{bb}$ | $AP^{bb}_r$ | $AP^{bb}_c$ | $AP^{bb}_f$ |
| End-to-end | 12 epochs | RFS [13] | *no* | 22.7 | 9.1 | 21.5 | 30.0 | 24.8 | 12.1 | 23.4 | 31.9 |
| | | | *yes* | **23.9** | **11.9** | **22.3** | **31.0** | **25.1** | **12.7** | **23.5** | **32.4** |
| | | IRFS [58] | *no* | 24.4 | 14.3 | 22.6 | 30.8 | 26.3 | 16.5 | 24.5 | 32.5 |
| | | | *yes* | **24.7** | **14.3** | **22.9** | **31.3** | **26.5** | **16.7** | **24.6** | **32.8** |
| | | EQLv2 [44] | *no* | 24.9 | 14.8 | 24.1 | 30.4 | 26.3 | 17.7 | 24.4 | 31.2 |
| | | | *yes* | **25.7** | **16.5** | **24.5** | **31.0** | **26.9** | **18.9** | **25.1** | **32.5** |
| | | SeeSaw [47] | *no* | 24.7 | 14.7 | 23.6 | 30.4 | 26.3 | 15.1 | 25.4 | 32.2 |
| | | | *yes* | **26.2** | **17.5** | **25.0** | **31.5** | **27.0** | **17.6** | **25.6** | **32.6** |
| | | ECM [24] | *no* | 26.5 | 17.0 | 25.4 | 31.7 | 27.9 | 19.2 | **26.5** | 33.5 |
| | | | *yes* | **27.3** | **19.2** | **25.9** | **32.5** | **28.0** | **19.5** | 26.2 | **33.7** |
| | | ROG [60] | *no* | 25.7 | 16.4 | 24.4 | 31.2 | 27.3 | 18.5 | 26.2 | 32.5 |
| | | | *yes* | **26.2** | **16.9** | **24.8** | **31.8** | **27.6** | **18.8** | **26.3** | **32.9** |
| Decoupled | 24+6 epochs | LOCE [11] | *no* | 27.2 | 18.7 | 25.7 | 32.6 | 28.5 | 19.0 | **27.0** | 34.3 |
| | | | *yes* | **27.6** | **18.9** | **26.5** | **33.0** | **28.7** | **20.2** | 26.8 | **34.4** |
| | 12+12 epochs | BACL [40] | *no* | 26.1 | 16.0 | 25.7 | 30.9 | 27.2 | 16.7 | 26.8 | 32.3 |
| | | | *yes* | **27.0** | **17.5** | **25.9** | **32.5** | **28.4** | **18.9** | **27.3** | **33.7** |

Table A.2: One-stage object detection results on LVIS v1.0 validation set. We compare different methods with ResNet-50 backbone on $2\times$ schedule using ATSS.

| Methods | $AP^{bb}$ | $AP^{bb}_r$ | $AP^{bb}_c$ | $AP^{bb}_f$ |
| --- | --- | --- | --- | --- |
| Focal Loss [34] | 25.6 | 14.5 | 24.3 | **31.8** |
| ECM [24] | 26.1 | 16.6 | 25.2 | 31.3 |
| Ours | **26.4** | **17.7** | **25.4** | 30.8 |

shown in Table A.1, using 2DRCL leads to consistent $AP^{bb}$ improvement over existing classification-based methods, surpassing all of them with large margins. Interestingly, combining our methods can be observed further growth in multiple paradigms. The method 'ECM+2DRCL' (which trained with a $1\times$ schedule) can almost achieve the same rare object detection accuracy as the LOCE [11] method, and surpasses BACL [40] for about 1.0% $AP^{bb}_r$. Therefore, we speculate that by using 2DRCL during training, the model can generate more balanced feature representations, allowing it to achieve comparable results to the decoupled method with minimal training when combined with end-to-end approaches.

**Comparison on ATSS Framework.** Table A.2 presents the performance comparison of our method against the baseline Focal Loss [34] and ECM Loss [24] on the ATSS [61] detection framework. Our method achieves the highest overall average precision ($AP^{bb}$) at 26.4%, outperforming both Focal Loss and ECM Loss. Notably, for rare classes, our method significantly improves performance with an $AP^{bb}_r$ of 17.7%, compared to 14.5% for Focal Loss and 16.6% for ECM Loss. Our method also shows consistent improvement for common classes, surpassing both Focal Loss and ECM Loss. Although Focal Loss achieves the highest precision for frequent classes, our method maintains competitive performance across all categories.

**Results on COCO-LT.** To further verify the generalization ability of our 2DRCL, we construct a long-tailed distribution dataset COCO-LT by sampling images and annotations from COCO [35] train 2017 split. Following [49], we divide 80 classes into 4 groups with < 20, 20-400, 400-8000, and >= 8000 training instances and report the accuracy for each group as $AP_1$, $AP_2$, $AP_3$, $AP_4$. In Table A.3, we compare our 2DRCL method with the baseline model and several state-of-the-art long-tailed detection methods on the COCO-LT dataset. The results demonstrate that 2DRCL consistently

Table A.3: Results on COCO-LT dataset. All experiments were conducted using the Mask R-CNN framework with a ResNet-50-FPN backbone and a $1\times$ training schedule.

| Methods | AP | $AP_1$ | $AP_2$ | $AP_3$ | $AP_4$ |
|---|---|---|---|---|---|
| CE | 18.7 | 0.0 | 8.2 | 24.4 | 26.0 |
| BAGS [29] | 21.5 | 13.4 | 17.7 | 22.5 | 26.0 |
| EQLv2 [44] | 23.1 | 3.8 | 17.4 | 25.8 | 29.4 |
| Seesaw [47] | 22.7 | 3.4 | 15.5 | **26.2** | 28.5 |
| ECM [24] | 22.9 | 11.0 | 18.7 | 25.7 | 28.7 |
| Ours | **24.4** | **14.4** | **20.2** | 26.1 | **29.4** |

Table A.4: Efficiency evaluation of Mask R-CNN with ResNet50-FPN.

| Methods | VRAM | Training Time | $AP^{bb}$ | $AP_r^{bb}$ |
|---|---|---|---|---|
| ECM (12 epochs) | **94 GB** | **16.1 h** | 26.5 | 17.0 |
| +AlignDet (6+6 epochs) | 103 GB | 19.5 h | 26.2 | 16.7 |
| +Ours (6+6 epochs) | 106 GB | 20.6 h | **27.0** | **19.0** |

outperforms the baseline model by a notable 5.7% in overall AP. Notably, 2DRCL achieves the highest AP across both group 1 and group 2, outperforming the closest competitor, ECM, by 3.4% and 1.5% AP, respectively. We attribute these improvements to 2DRCL's ability to mitigate the simplicity bias toward tail classes during pre-training, which contrasts with methods such as Seesaw and ECM that primarily address the issue of unequal competition between foreground classes without sufficiently addressing the inherent bias in representation learning. By directly confronting these biases, our method demonstrates substantial gains in tail category performance while maintaining strong results across the full distribution.

**Efficiency Evaluations.**   Table A.4 compares the VRAM usage, training time, and performance of different methods for long-tailed object detection. Our method, which includes 6 epochs of pre-training followed by 6 epochs of fine-tuning, utilizes slightly more VRAM and training time compared to AlignDet and ECM. Despite the increased computational cost, our method delivers a notable performance boost, achieving the highest $AP^{bb}$ of 27.0% and $AP_r^{bb}$ of 19.0%, demonstrating a clear performance-cost trade-off. These results indicate that while our pre-training strategy demands more resources, it does not negatively impact fine-tuning performance and offers significant improvements in long-tailed detection.

**Qualitative results.**   We present additional visualization results on the LVIS dataset [13]. For simplicity, we use RFS [13] with Cross-Entropy (CE) loss as the baseline and combine it with our 2DRCL method. As shown in Figure A.1, both the baseline and our 2DRCL detect most objects in an image, but 2DRCL generally captures more details. For instance, 2DRCL helps discover missed boxes, such as the horse carriage in the center-left images. Additionally, 2DRCL generates more accurate bounding box predictions, like correctly identifying boats on the ground. While using a basic method still results in numerous classification errors, this issue can be mitigated by carefully designing the loss function.

**Bias Analyses.**   We visualize how 2DRCL mitigates the weight norm bias induced by long-tailed distributions. Figure A.2a shows the classifier weight norm distribution across classes for models trained with the LVIS v1.0 training split. The transparent lines represent actual weight norms, while the solid lines show polynomial fits, providing a smoothed interpretation of trends. Our 2DRCL method achieves weight norm performance comparable to ECM [24], indicating effective bias mitigation. Figures A.2b and A.2c further illustrate that each component of 2DRCL contributes to reducing weight norm bias. Despite these gains, some advantages achieved at the feature level are slightly diminished during downstream fine-tuning, limiting overall progress in final outcomes.

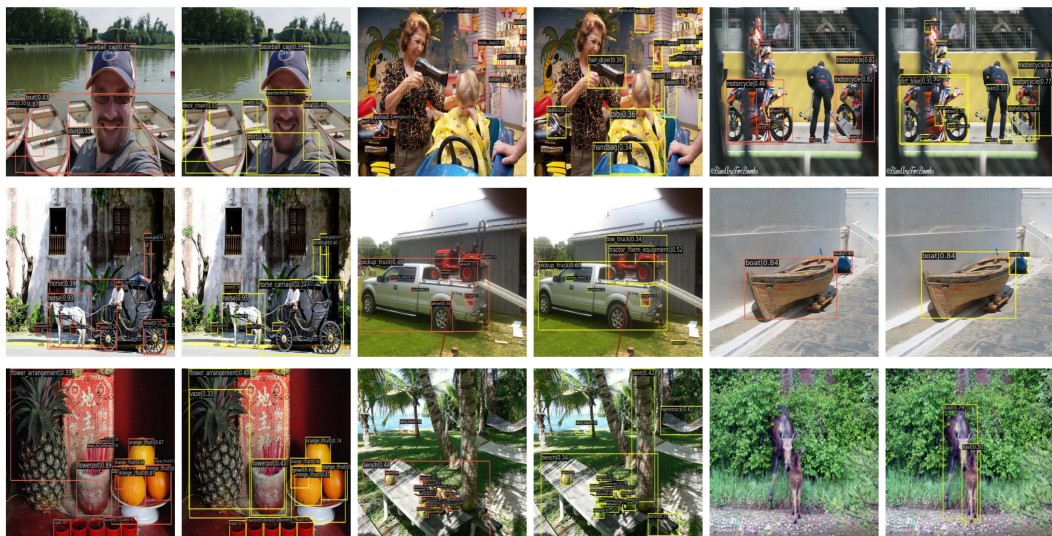

Figure A.1: Visualizations of detection results before (in the left of each group) and after (in the right) using our 2DRCL. We adopted RFS [13] as the baseline in LVIS and combined it with our 2DRCL pre-training method. In comparison, the proposed method is good at detecting missing objects and rectifying bounding box predictions. This figure needs to be viewed in color.

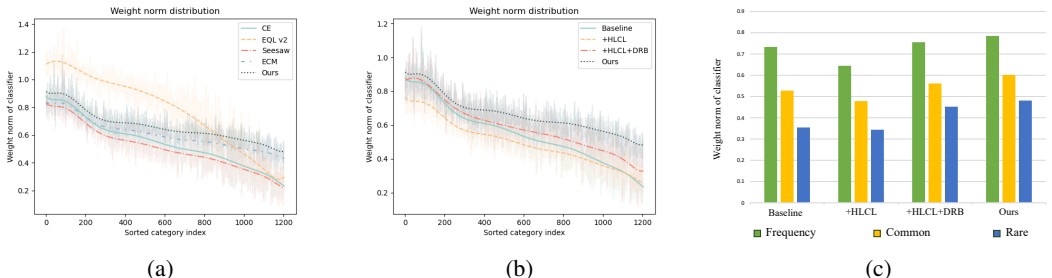

(a)                    (b)                    (c)

Figure A.2: (a) and (b) are classifiers' weight norm distribution across different classes in Mask R-CNN models trained with the LVIS v1.0 training split [13]. The X-axis represents the sorted category index based on category frequency. The Y-axis shows the weight norm. Transparent lines depict the actual weight norms for each category, providing a raw look at the data distribution. The solid lines represent polynomial curves fitted to the transparent data, offering a smoothed interpretation of trends across classes. (a) represents the comparison with the state-of-the-art methods, while (b) represents the comparison with the proposed components in this paper. (c) represents the average weight norm of the classifiers for each frequency category.

