# OpenReview forum: "Long-tailed Object Detection Pretraining: Dynamic Rebalancing Contrastive Learning with Dual Reconstruction"
_NeurIPS.cc/2024/Conference — NeurIPS 2024 poster_

### Official Review · Reviewer_ZXbG · 2024-07-04

**Soundness:** 3
**Presentation:** 3
**Contribution:** 3
**Rating:** 6
**Confidence:** 3

**Summary:**

This paper proposes Dynamic Rebalancing Contrastive Learning with Dual Reconstruction (DRCL) to tackle long-tailed object detection challenges. DRCL integrates dynamic rebalancing to address instance-level imbalance and a dual reconstruction strategy to enhance feature representation for tail categories. Experimental results on the LVIS v1.0 dataset demonstrate that DRCL achieves State-of-the-Art performance, significantly improving detection accuracy for rare categories across various detectors and backbones. The method shows an average improvement of 0.9% APb, with substantial gains in rare categories (+1.5% APbr), highlighting its effectiveness and competitiveness.

**Strengths:**

+ This paper is well-written and clearly structured, making it easy to follow the methodology and understand the contributions. The clarity in presentation enhances the readability and ensures that the proposed concepts and findings are effectively communicated.

+ The motivation behind the proposed method is strong, focusing on the critical issue of long-tailed object detection. The design of the modules within the method is well-targeted and thoughtfully crafted to address this specific problem, showing a deep understanding of the challenges involved.

+ The experimental results are robust, demonstrating significant improvements in performance. The method's versatility is highlighted by its applicability across various detectors and backbones, reinforcing its potential for widespread use in different settings.

**Weaknesses:**

One weakness of the paper is that some section titles could benefit from further refinement. For instance, the titles of sections 3.1.1, 3.1.2, and 4.3 may not fully capture the content they encompass, potentially causing confusion for readers. Improving the clarity and descriptiveness of these section names would enhance the overall structure and navigability of the paper.

**Questions:**

Figure 4 brings more insightful information. However, more discussions are encouraged to be involved in the corresponding content.

---

> ### Author Rebuttal · Authors · 2024-08-06
>
> ## Thank you for the positive comments. Below please find our point-to-point responses.
>
> *Comment_1: One weakness of the paper is that some section titles could benefit from further refinement. For instance, the titles of sections 3.1.1, 3.1.2, and 4.3 may not fully capture the content they encompass, potentially causing confusion for readers. Improving the clarity and descriptiveness of these section names would enhance the overall structure and navigability of the paper.*
>
> Response_1: In the final version of the paper, we will improve the clarity and descriptiveness of these section titles to enhance the overall structure and navigability of the paper. This will help ensure that the content is clearly communicated and easily understood.
>
> ---
>
> *Comment_2: Figure 4 brings more insightful information. However, more discussions are encouraged to be involved in the corresponding content.*
>
> Response_2: In the final version of the paper, we will include a deeper analysis of the various types of errors presented in Figure 4, combined with more extensive data. This will provide a more comprehensive discussion and insights into the conditions under which our method performs well and where it may need improvement.

---

### Official Review · Reviewer_XbTc · 2024-07-11

**Soundness:** 2
**Presentation:** 3
**Contribution:** 2
**Rating:** 6
**Confidence:** 4

**Summary:**

The paper proposes DRCL, an object detection pretraining methodology for datasets with long-tailed object class distributions. Their proposed framework consists of three losses: 1) image-level constrastive instance discrimination, 2) object-level contrastive instance discrimination, and 3) a reconstruction loss, with both pixel-wise and feature-wise components. Furthermore, the paper proposes a rebalancing sampling method that takes into account both image count and instance count when estimating the repeat factor per image. The results indicate that the proposed pretraining method is beneficial for long-tailed object detection.

**Strengths:**

**1)** The proposed framework is intuitive and clearly presented.

**2)** Leveraging self-supervised methodologies for detection pretraining is a proven approach. The proposed method specifically is interesting in that it demonstrates that different self-supervised objectives (image-level contrast, object-level contrast, class-unaware detection and reconstruction) can be effectively incorporated in a single pipeline.

**3)** The proposed pretraining is beneficial, as demonstrated by the paper's results.

**4)** The problem tackled by the paper (i.e. tackling datasets with imbalanced classes) is significant in the object detection domain.

**Weaknesses:**

**1)** The proposed method is not self-supervised, as the dynamic rebalancing component implies and requires object label information for the images of the training set. Specifically, we need to know in advance how many instances of each class are present in each specific image. That should be clear in the paper, as there is phrasing that implies the proposed method is self-supervised (L35-36, L226-230).

**2)** I believe there are several issues with the evaluations conducted and the results presented:
\
**2A)** The comparisons of Table 1 are unfair. Compared to the baseline long-tailed frameworks the proposed method requires twice as much training (12 pretraining epochs on top of the 12 finetuning epochs). Therefore, that it leads to performance gains is to be expected. A fair comparison would require roughly equal computational budget. Succinctly: since pretraining and fine-tuning are done on the same dataset, the key question is not whether additional training with the proposed method is beneficial (which is what is shown in Table 1 and is trivial), but if the same computational budget is better spent pretraining and finetuning instead of simply finetuning for longer with a baseline long-tailed framework.
\
**2B)** The comparisons in Table 3 are, in my opinion, not meaningful. Regarding SoCo, it pretrains the backbone from scratch, whereas AlignDet and the proposed method use a pretrained (indeed supervised) backbone. More broadly, they are fully self-supervised methods whereas the proposed method leverages label information. For a fair comparison, the proposed method should be evaluated without the dynamic rebalancing component, with the same pretraining data, and taking into account the computational cost of pretraining.
\
**2C)** The pretraining time and VRAM requirements of the method should be included in the paper. If I understand correctly, the proposed method requires (for each training step) 3 backbone forward passes (2x encoder + 1x momentum encoder), 2 passes from the detection head, and 1 decoder pass. The computational cost is likely much greater than the baseline long-tailed methods and self-supervised methods like AlignDet, a fact which should be taken into account when contrasting the results presented.
\
**2D)** The proposed method is only applied to one dataset (for both pretraining and finetuning) and with only one type of detector (Faster/Mask R-CNN).

**3)** The novelty of the proposed method is, in my opinion, somewhat limited. In effect, it is a combination of MoCo (image level instance discrimination), SoCo (object level instance discrimination), and masked image modelling methods (pixel/feature reconstruction).  The dynamic rebalancing method is a minor improvement over [11]. Overall, while the effective combination of these methods does represent a meaningful contribution, I am not sure it is up to the level of NeurIPS. I also consider important in this regard the framework's efficiency: if the authors demonstrated that they have combined these approaches while keeping the framework relatively efficient (compared to the original approaches) the impact of the proposed method would be significantly greater.

**4)** The paper is, in places, poorly written and imprecise:
\
**4A)** Abstract: "By synergistically combining self12 supervised and supervised learning modalities, our approach substantially reduces pretraining time and resource demands." I think the term modality is used incorrectly in this context, and nowhere in the paper is it claimed and/or supported that the proposed method is more efficient that the alternatives. Indeed, the opposite is mentioned in the conclusion (L291).
\
**4B)** Sec. 3.2 "training" should be replaced with "pre-training" to make clear that this applies to the pretraining stage, not the downstream task fine-tuning (training) stage, following the distinction the authors make in Appendix Sec. 1.
\
**4C)** L189: "We combine our DRCL with existing long-tailed methods". I assume the authors mean they pretrain with their method and fine-tune with the corresponding long-tailed methods. This should be stated clearly to avoid confusion.
\
**4D)** Pretraining/training epochs: In L199-202, L431, L437, and L447 references are made to pretraining and training epochs. This information should be condensed and made clearer to avoid confusion between training and pretraining.
\
**4E)** It is not mentioned which class-agnostic object detector is used to extract object proposals for detection pretraining, what is the architecture of the generator, and the width of the projection/predictor MLPs.
\
**4F)** L240 mentions that SoCo is pretrained for 530 epochs. This is misleading without the clarification that it pretrains the backbone from scratch, whereas the proposed method and AlignDet use pretrained backbones. Additionally, it is my understanding that AlignDet and the proposed method are pretrained for approximately the same number of steps (AlignDet pretrains for 12 epochs on COCO's train set). Therefore, the "despite their extensive training epochs" comment in L242 is misleading.

**5F)** The related works section is very limited. It should properly reference at least the most established and relevant self-supervised methods for backbone (MoCo, SwAV, DINO, SimCLR etc. for object-centric and SoCo, SlotCon, Odin, DetCon etc. for scene-centric pretraining) and for detector pretraining (Up-DETR, DETReg, CutLER etc.). Furthermore, it is unclear what is meant by the fact that AlignDet "decouples" pretraining to avoid high costs. Similar to the proposed method, AlignDet uses a pretrained backbone, the only distinction being that the backbone is frozen during detection pretraining. If that is what is meant, it should be made clear.

**Questions:**

In my opinion, the most important issue is related to the experiments, based on my understanding that the method is not self-supervised and that the results in Table 1 compare "12 ep. DRCL pretraining + 12 ep. long-tailed finetuning" with "12 ep. long-tailed finetuning". If that is indeed the case, I believe different experiments are required to support the claims of the paper. Specifically, as mentioned in the weaknesses, I would suggest that the authors include in the paper:
\
**a)** the computational cost (Time/epoch & VRAM requirements) of their method compared to the most significant alternative works (supervised long-tailed and self-supervised),
\
**b)** the results of finetuning baseline long-tailed methods for the same number of epochs (or a computationally fair equivalent) as the proposed method (including the finetuning).
\
Overall, as I mentioned in the weaknesses, given that the proposed method requires some label information, to demonstrate its utility the authors must, in my opinion, demonstrate that it is worth pretraining and then finetuning, rather than simply finetuning for longer (or even with a bigger model given the, I assume, increased VRAM requirements of DRCL).

Regarding the novelty issue I raised in weakness 3, I would welcome the authors' highlighting aspects I might have missed or underestimated regarding the novelty in their proposed approach.

I emphasize, however, that my main concern with the paper relates to the (in my opinion) insufficient evaluations, and that the writting issues and the novelty concern are secondary.

**Limitations:**

The authors adequately addressed the limitations.

---

> ### Author Rebuttal · Authors · 2024-08-06
>
> ## Thanks for the comments.
>
> Comment_1: The proposed method is not self-supervised.
>
> Response_1: The self-supervised components mentioned in our paper refer specifically to the Holistic-Object Contrastive Learning and Dual Reconstruction training modes. In the final version, we will eliminate any ambiguous statements and make it clear that our method involves supervised elements due to the dynamic rebalancing component requiring object label information.
>
> Comment_2A: The comparisons of Table 1 are unfair.
>
> Response_2A: To ensure fairness, we first employed AlignDet, one of the best current pretraining frameworks for object detection, to perform 12 epochs of pretraining on the LVIS dataset. Subsequently, we used a selection of representative long-tailed object detection methods for comparison. The results showed that our method provides significant improvements.
>
> | Methods | $\mathrm{AP}^{b}$ | $\mathrm{AP}_{r}^{b}$ | $\mathrm{AP}_{c}^{b}$ | $\mathrm{AP}_{f}^{b}$ |
> |---|---|---|---|---|
> | AlignDet+RFS | 22.8 | 10.1 | 21.2 | 30.2 |
> | Ours+RFS | **23.9** | **11.9** | **22.3** | **31.0** |
> |
> | AlignDet+Seesaw | 25.2 | 15.3 | 23.5 | 30.9 |
> | Ours+Seesaw | **26.2** | **17.5** | **25.0** | **31.5** |
> |
> | AlignDet+ECM | 26.5 | 17.9 | 25.2 | 31.6 |
> | Ours+ECM | **27.3** | **19.2** | **25.9** | **32.5** |
> |
> | AlignDet+ROG | 25.6 | 16.1 | 24.6 | 31.1 |
> | Ours+ROG | **26.2** | **16.9** | **24.8** | **31.8** |
>
> Comment_2B: The comparisons in Table 3 are not meaningful.
>
> Response_2B: Following your suggestion, we conduct additional experiments with a modified setup without the dynamic rebalancing component and with the same pretraining data. Specifically, we replaced the from-scratch backbone of SoCo with a pretrained backbone, denoted as SoCo*, and performed the same pretraining on the LVIS dataset. The experimental results show that AlignDet outperforms SoCo*, and our method surpasses AlignDet. This demonstrates the effectiveness of our approach.
>
> | Methods | $\mathrm{AP}^{b}$ | $\mathrm{AP}_{r}^{b}$ | $\mathrm{AP}_{c}^{b}$ | $\mathrm{AP}_{f}^{b}$ |
> |---|---|---|---|---|
> | SoCo* | 22.4 | 8.9 | 20.1 | 30.3 |
> | AlignDet | 22.8 | 10.1 | 21.2 | 30.2 |
> | Ours | **23.7** | **11.0** | **21.9** | **31.1** |
>
> AlignDet is one of the most advanced pretraining schemes for object detection and has been compared with SoCo, highlighting SoCo as a strong competitor. By comparing our method with them on the LVIS dataset, we aim to demonstrate that our method is more suitable for handling long-tailed data.
>
> Comment_2C: The pretraining time and VRAM requirements of the method should be included in the paper.
>
> Response_2C: We will include this information in the final version of the paper to ensure a comprehensive understanding of the computational resources involved.
>
> Comment_2D: The proposed method is only applied to one dataset and with only one type of detector.
>
> Response_2D: Following your suggestion, we further conduct experiments on the COCO-LT dataset, and report the results in the following table. As shown, the proposed method demonstrates a significant advantage.
>
> | Method | $\mathrm{AP}$ | $\mathrm{AP}_{1}$ | $\mathrm{AP}_{2}$ | $\mathrm{AP}_{3}$ | $\mathrm{AP}_{4}$ |
> |---|---|---|---|---|---|
> | Base | 18.7 | 0.0 | 8.2 | 24.4 | 26.0 |
> | EQLv2 | 23.1 | 3.8 | 17.4 | 25.8 | **29.4** |
> | Seesaw | 22.9 | 3.4 | 15.5 | **26.2** | 28.5 |
> | ECM | 22.7 | 11.0 | 18.7 | 25.7 | 28.7 |
> | BAGS | 21.5 | 13.4 | 17.7 | 22.5 | 26.0 |
> | Ours | **24.4** | **14.4** | **20.2** | 26.1 | **29.4** |
>
> Additionally, we conduct experiments with another type of detector, i.e., ATSS [Ref 1], and report the results in the following table. As shown, our method achieves the best results.
>
> | Methods | $\mathrm{AP}^{b}$ | $\mathrm{AP}_{r}^{b}$ | $\mathrm{AP}_{c}^{b}$ | $\mathrm{AP}_{f}^{b}$ |
> |---|---|---|---|---|
> | Focal Loss | 25.6 | 14.5 | 24.3 | **31.8** |
> | ECM | 26.1 | 16.6 | 25.2 | 31.3 |
> | Ours | **26.4** | **17.7** | **25.4** | 30.8 |
>
> [Ref 1] Bridging the gap between anchor-based and anchor-free detection via adaptive training sample selection, CVPR, 2020.
>
>
> Comment_3: The novelty of the proposed method is limited.
>
> Response_3: We would like to emphasize that we inspect long-tailed object detection through the lens of simplicity bias (SB). As described in L138-148 and observed in Figure 3, we found that tail classes in long-tailed object detection suffer significantly from SB, a problem that has not been adequately explored by existing methods. Based on this observation and mechanism, we specifically proposed Dual Reconstruction to tackle this problem. As illustrated in Figure 3 of the paper, the inclusion of Dual Reconstruction significantly enriches the feature representation of tail classes and alleviates the SB problem. This provides new insight into the field.
>
> Additionally, the dynamic resampling strategy is designed explicitly for long-tailed object detection. These contributions highlight the novelty and targeted nature of our approach. As highlighted by Reviewer ZXbG in Strengths_2, the modules in our proposed method are specifically tailored for long-tailed object detection.
>
> Simply combining existing methods would result in significant resource and time consumption, with a pretraining speed of 2.37 s/iter. Through our efficient implementation of a unified framework, we have optimized GPU utilization by employing mixed-precision training and limiting the number of generated bounding boxes used for pretraining to a maximum of 8 per image. These optimizations have reduced the pretraining speed to 0.96 s/iter. It ensures the practicality of our method without requiring more powerful hardware resources.
>
> Comment_4: The paper is, in places, poorly written and imprecise. The related works section is very limited.
>
> Reponse_4: We will thoroughly review the entire paper to address these issues, making necessary corrections and additions in the final version. If needed, we can provide specific details during the discussion stage.

---

> > ### Comment · Reviewer_XbTc · 2024-08-08
> > **Reply to authors**
> >
> > I thank the authors for their effort to respond to the issues I raised in my review.
> > However, they have not addressed my main concerns regarding their work.
> >
> > As it stands, their work is presented as a **supervised** pretraining method, to be applied on the same data as the finetuning. As such, in my opinion, the main comparison should be with computationally equivalent supervised finetuning (12 ep pretraining + 12 ep finetuning vs 24 ep finetuning) as I outlined in my review. Instead, the authors presented in the rebuttal comparisons with self-supervised pretraining.
> >
> > Such a comparison would be more significant if the paper was presented as a self-supervised pretraining method. However, even in that case, fair comparisons should take into account the computational cost of each method. That is the second major issue my review raised and the authors did not address.
> >
> > Given all the other experiments that the authors were able to run for the rebuttal, I would have expected a comparison regarding the VRAM and training time requirements of the proposed method to have been included in the rebuttal. The computational cost of a method is critical toward evaluating it and, as such, I do not believe it can be left for the final version post-acceptance: it should be considered **prior** to acceptance. Especially when, as I outlined in my review, the proposed method appears to be much more costly than its competitors. The performance of the proposed method is promising (especially considering the results presented in the rebuttal), but we as reviewers should be aware of the performance-cost trade-off to be able to evaluate the method.
> >
> > I hope that the authors will take advantage of the remaining time in the discussion period to produce such results. I want to emphasize that a vram/training time contrast should be done on equal terms (i.e. with/without mixed precision for all methods).
> >
> > As I stated in my original review, in my opinion, the authors should produce: a) a fair VRAM and training time comparison with AlignDet and, ideally, a supervised finetuning method, and b) results for 24 ep finetuning with at least one strong finetuning baseline (or a roughly computationally equivalent number of epochs if the training cost requirements are too large).

---

> > > ### Author Response · Authors · 2024-08-12
> > > **Reply to Reviewer XbTc**
> > >
> > > Thank you for the further discussions. Following your constructive suggestions, we conducted additional experiments as outlined below.
> > >
> > > Regarding point (a), we performed a VRAM and training time comparison with AlignDet and ECM (a supervised fine-tuning method). As shown in the following table, our method achieves the best results for long-tailed detection tasks. Additionally, the VRAM usage and training time of our method are not significantly larger than those of AlignDet and ECM, making it acceptable in practice.
> > >
> > > | Methods | VRAM | Training Time | $\mathrm{AP}^{b}$ | $\mathrm{AP}_{r}^{b}$ |
> > > |:---:|:---:|:---:|:---:|:---:|
> > > | ECM (12 epochs) | **94 GB** | **16.1 h** | 26.5 | 17.0 |
> > > | +AlignDet (6+6 epochs) | 103 GB | 19.5 h | 26.2 | 16.7 |
> > > | +Ours (6+6 epochs) | 106 GB | 20.6 h | **27.0** | **19.0** |
> > >
> > > Regarding point (b), we performed fair comparisons as you suggested. Due to the large training cost requirements, we compared results after a total of 12 epochs. As reported in the following table, our method achieves an average 0.5% accuracy improvement over the baselines, which is significant for long-tailed object detection tasks.
> > >
> > > | Methods | $\mathrm{AP}^{b}$ | $\mathrm{AP}_{r}^{b}$ | $\mathrm{AP}_{c}^{b}$ | $\mathrm{AP}_{f}^{b}$ |
> > > |:---:|:---:|:---:|:---:|:---:|
> > > | RFS (12 epochs) | 22.7 | 9.1 | 21.5 | 30.0 |
> > > | +Ours (6+6 epochs) | **23.3**  | **10.3** | **21.7** | **30.3** |
> > > | |
> > > | EQL (12 epochs) | 24.9 | 14.8 | 24.1 | **30.4** |
> > > | +Ours (6+6 epochs) | **25.2** | **15.9** | **24.3** | 30.3 |
> > > | |
> > > | Seesaw (12 epochs) | 24.7 | 14.7 | 23.6 | 30.4 |
> > > | +Ours (6+6 epochs) | **25.2** | **15.6** | **24.2** | **30.5** |
> > > | |
> > > | ECM (12 epochs) | 26.5 | 17.0 | 25.4 | 31.7 |
> > > | +Ours (6+6 epochs) | **27.0** | **19.0** | **25.8** | **31.9** |

---

> > > > ### Comment · Reviewer_XbTc · 2024-08-12
> > > > **Reply to authors**
> > > >
> > > > I thank the authors for replying to my comment and for providing these additional experiments.
> > > >
> > > > I am very satisfied with the results the proposed method achieves, particularly with the relatively low VRAM and training time costs.
> > > >
> > > > I am still a little concerned about how these costs (+30% training time) would scale to larger and better architectures (e.g. DETR-based detectors), and about the fact that I feel this method would be much more impactful if it was primarily focused on self-supervised detector pretraining. However, in light of the author's response, I am happy to say that I am inclined to raise my recommendation to Weak Accept (6).

---

> > > > > ### Author Response · Authors · 2024-08-13
> > > > > **Reply to Reviewer XbTc**
> > > > >
> > > > > Thank you very much for reconsidering our work and for your updated feedback. We sincerely appreciate your thoughtful evaluation and the change in your recommendation to a weak accept. Your constructive comments have been invaluable in improving our submission.

---

### Official Review · Reviewer_gQpU · 2024-07-11

**Soundness:** 4
**Presentation:** 4
**Contribution:** 4
**Rating:** 7
**Confidence:** 5

**Summary:**

This paper tackles the underperformance of object detection on long-tailed datasets using a novel pretraining methodology called Dynamic Rebalancing Contrastive Learning with Dual Reconstruction (DRCL). DRCL addresses biases in classifier weight norms and feature representation by integrating holistic and object-level contrasts, employing dynamic rebalancing from image-level to instance-level resampling, and maintaining both natural appearance and semantic consistency. Combining self-supervised and supervised learning, DRCL reduces pretraining time and resources, achieving state-of-the-art performance on the LVIS dataset across multiple detection frameworks and backbone networks.

**Strengths:**

1. **Clear Presentation:** The paper is well-presented and easy to follow, ensuring that the research is accessible and comprehensible.

2. **Practical and Valuable Research:** The focus on long-tailed object detection is highly practical and valuable, addressing a significant challenge in the field. The motivation behind the proposed methodology is logical and well-founded.

3. **Innovative Methodology:** The approach demonstrates considerable innovation, and the findings, such as those illustrated in Figure 2, effectively support the motivation and design of the method.

4. **Effective Experimental Validation:** The experimental results robustly validate the effectiveness of the proposed method, demonstrating its superiority in handling long-tailed distributions in object detection.

**Weaknesses:**

1. The paper lacks qualitative analysis, which could provide a more in-depth understanding of the model's performance and behavior.

2. There is an absence of specific failure case analysis, making it difficult to understand the conditions under which the proposed method may not perform well.

**Questions:**

What are the specific values of the trade-off parameters in Equation 9?

---

> ### Author Rebuttal · Authors · 2024-08-06
>
> ## Thank you for the positive comments. Below please find our point-to-point responses.
>
> *Comment_1: The paper lacks qualitative analysis, which could provide a more in-depth understanding of the model's performance and behavior.*
>
> Response_1: We have already included analyses in our paper with Figure 5, presenting visualized results of the detected bounding boxes, Figure 3, showcasing feature activation maps to highlight the advantages of our method, and Figure 4, presenting an error analysis. In the final version, we will conduct a deeper analysis and consider providing additional qualitative analyses, such as more detailed error analysis, class-specific performance examples. These additions will provide a comprehensive understanding of our model's strengths and areas for improvement.
>
> ---
>
> *Comment_2: There is an absence of specific failure case analysis, making it difficult to understand the conditions under which the proposed method may not perform well.*
>
> Response_2: We will address it by including a detailed analysis of failure cases in the final version of our paper. This will help in understanding the limitations of our method and provide insights for further improvements.
>
> ---
>
> *Comment_3: What are the specific values of the trade-off parameters in Equation 9?*
>
> Response_3: In our experiments, the trade-off parameters in Equation 9 for \mathcal{L}_{HOC}, \mathcal{L}_{DRC} and \mathcal{L}_{det} were all set to 1. We will ensure this information is clearly stated in the final version of the paper.

---

### Official Review · Reviewer_kReB · 2024-07-18

**Soundness:** 2
**Presentation:** 3
**Contribution:** 2
**Rating:** 5
**Confidence:** 3

**Summary:**

The authors proposed a pre-training method for long-tail object detection. Specifically, the authors integrated holistic and object-level contrast within a contrastive learning framework, used a dynamic rebalancing technique to transition from image-level resampling to instance-level resampling, and implemented a dual reconstruction strategy to maintain natural appearance and internal semantic consistency. Extensive experiments on LVIS demonstrated its effectiveness.

**Strengths:**

1) The data in the world follows the long-tailed distribution and it is meaningful to solve the imbalanced object detection problem.
2) The paper is well organized.
3) The proposed method can be combined with different existing methods.

**Weaknesses:**

1) The long-tailed object detection problem has already been explored, the author should give more discussion about the existing works and the proposed method.
2) The proposed method is too complex and involves different losses to supervise the learning. The author should give more analysis of these losses and discuss their relationship.
3) The author should evaluate the method on more datasets, like COCO-LT.
4) Too many hyper-parameters and how to set them is confusing.
5) The author compares the proposed method to different self-supervised learning methods. However, I think traditional methods like MOCO,  SimCLR, etc are potential baselines and should also be compared.

**Questions:**

see weakness

**Limitations:**

yes

---

> ### Author Rebuttal · Authors · 2024-08-06
>
> ## Thank you for the comments. Below please find our responses to some specific comments.
>
> *Comment_1: The author should give more discussion about the existing works and the proposed method.*
>
> Response_1: Our proposed method addresses a significant gap in long-tailed object detection by introducing a pretraining strategy that specifically mitigates the simplicity bias in underrepresented tail classes through a component designed for this purpose (DRCL). This approach effectively addresses the insufficiency in tail class feature representation seen in current methods. By emphasizing tail class features, our method integrates seamlessly with existing long-tailed detection approaches, resulting in improved detection performance. We will further include a more detailed discussion of related work and the unique contributions of our method in the revised version of the paper.
>
> ---
>
> *Comment_2: The author should give more analysis of these losses and discuss their relationship.*
>
> Response_2: In the paper, we have indeed conducted a detailed ablation study of these losses in Table 4. Furthermore, we also supplement this with an ablation study specifically for the Holistic-Object Contrastive (HOC) component.
>
> | Holistic-level Contrast | Object-level Contrast | DRB | AR | SR | $\mathrm{AP}^{b}$ | $\mathrm{AP}_{r}^{b}$ | $\mathrm{AP}_{c}^{b}$ | $\mathrm{AP}_{f}^{b}$ |
> |---|---|---|---|---|---|---|---|---|
> | × | × | × | × | × | 22.7 | 9.1 | 21.5 | 30.0 |
> | √ | × | × | × | × | 22.5 | 10.5 | 21.0 | 29.3 |
> | × | √ | × | × | × | 21.9 | 9.8 | 20.8 | 28.7 |
> | √ | √ | × | × | × | 22.4 | 10.8 | 21.1 | 29.0 |
> | √ | √ | √ | × | × | 23.8 | 14.3 | 22.3 | 30.1 |
> | √ | √ | √ | √ | × | 24.2 | 14.9 | 22.6 | 30.3 |
> | √ | √ | √ | √ | √ | **24.4** | **15.2** | **22.7** | **30.3** |
>
> From the results presented in the table, it is evident that both the Holistic-level Contrast and Object-level Contrast individually contribute to improved performance on tail classes. The Holistic-level Contrast drives the model to learn global differences and similarities across images, enhancing overall feature learning. The Object-level Contrast focuses on capturing local differences and similarities, refining features crucial for object detection tasks.
>
> The Dual Reconstruction approach aims to enrich the feature representation of tail classes. The Appearance Consistency Reconstruction focuses on pixel-level reconstruction. Furthermore, we introduce Semantic Consistency Reconstruction, which ensures that the model pays more attention to semantic consistency rather than just pixel-wise details. This enhances the model's robustness by promoting semantic-level understanding and reducing overfitting risks. We will include these results in the revised paper.
>
> ---
>
> *Comment_3: The author should evaluate the method on more datasets, like COCO-LT.*
>
> Response_3: We supplement our experiments with evaluations on the COCO-LT dataset. As shown in the table, our approach outperforms existing methods with an overall average AP improvement of 1.3%. It demonstrates a significant advantage, particularly in tail classes.
>
> | Method | $\mathrm{AP}$ | $\mathrm{AP}_{1}$ | $\mathrm{AP}_{2}$ | $\mathrm{AP}_{3}$ | $\mathrm{AP}_{4}$ |
> |---|---|---|---|---|---|
> | Base | 18.7 | 0.0 | 8.2 | 24.4 | 26.0 |
> | EQLv2 | 23.1 | 3.8 | 17.4 | 25.8 | **29.4** |
> | Seesaw | 22.9 | 3.4 | 15.5 | **26.2** | 28.5 |
> | ECM | 22.7 | 11.0 | 18.7 | 25.7 | 28.7 |
> | BAGS | 21.5 | 13.4 | 17.7 | 22.5 | 26.0 |
> | Ours | **24.4** | **14.4** | **20.2** | 26.1 | **29.4** |
>
> ---
>
> *Comment_4: Too many hyper-parameters and how to set them is confusing.*
>
> Response_4: We used the default settings (cf. L113, L130, L183) as provided in the paper for all experiments, without extensive hyper-parameter tuning. This demonstrates the robustness of our method and its insensitivity to parameter changes. In fact, tuning the hyper-parameters is straightforward. It can be accomplished by dividing the dataset to include a validation set and systematically adjusting the parameters.
>
> ---
>
> *Comment_5: The author compares the proposed method to different self-supervised learning methods. However, I think traditional methods like MOCO, SimCLR, etc are potential baselines and should also be compared.*
>
> Response_5: We conduct additional experiments with traditional methods such as MOCO, SimCLR, and BYOL, following your suggestions for comparison. We will include these results and their analysis in our final version to provide a comprehensive comparison.
>
> | Method | $\mathrm{AP}^{b}$ | $\mathrm{AP}_{r}^{b}$ | $\mathrm{AP}_{c}^{b}$ | $\mathrm{AP}_{f}^{b}$ |
> |---|---|---|---|---|
> | MoCo v3 | 14.5 | 3.9 | 12.4 | 21.6 |
> | SimCLR | 19.9 | 8.0 | 18.1 | 27.1 |
> | BYOL | 15.3 | 5.4 | 13.2 | 21.9 |
> | Ours | **23.9** | **11.9** | **22.3** | **31.0** |

---

> > ### Author Response · Authors · 2024-08-13
> >
> > Dear Reviewer kReB,
> >
> > Thank you again for your valuable comments. We are eager to know if our responses have addressed your concerns and look forward to your further feedback. Please feel free to reach out with any additional questions.
> >
> > Best regards,
> >
> > The Authors

---

> > > ### Comment · Reviewer_kReB · 2024-08-13
> > >
> > > Thanks for your response, after reading the responses and other reviewers' comments, I decide to raise my score.

---

### Decision · Program_Chairs · 2024-09-25

**Decision:**

Accept (poster)

**Comment:**

The paper explores long-tailed object detection, introducing Dynamic Rebalancing Contrastive Learning with Dual Reconstruction (DRCL) to mitigate biases. Initially, reviewers expressed concerns about the approach's complexity and the lack of sufficient experimental results. However, the authors successfully addressed these issues in their rebuttal, leading to unanimous positive feedback from the reviewers. After reviewing the comments and rebuttal, the AC recommends acceptance. The authors are encouraged to incorporate the feedback, particularly concerning the discussion and connection to prior work, into the final camera-ready version.